# Wellness Impacts of Social Capital Built in Online Peer Support Forums

**DOI:** 10.3390/ijerph192315427

**Published:** 2022-11-22

**Authors:** Sue Kilpatrick, Sherridan Emery, Jane Farmer, Peter Kamstra

**Affiliations:** 1College of Arts, Law and Education, University of Tasmania, Hobart, TAS 7005, Australia; 2Social Innovation Research Institute, Swinburne University of Technology, Hawthorn, VIC 3122, Australia

**Keywords:** rural mental health, social capital, online peer support forums, reciprocity

## Abstract

The study reported in this paper sought to explore whether and how social capital resources were generated on online peer support mental health forums, and how they were used by rural users to influence mental health outcomes. Interviews with rural users of three Australian online peer support mental health forums were analysed to identify interactions that accessed social capital resources and mental wellness outcomes that flowed from these. Analysis drew on a model of simultaneous building and using of social capital to uncover the nature of the social capital resources present on the forum and how they were built. Findings show that forums were sites for building ‘knowledge resources’ including archives of users’ experiences of navigating mental illness and the mental health service system; and ‘identity resources’ including a willingness to contribute in line with forum values. The knowledge and identity resources built and available to rural users on the forums are facilitated by forum characteristics, which can be viewed as affordances of technology and institutional affordances. Operation by trusted organisations, moderation, a large network of users and anonymity created a safe space that encouraged reciprocity and where users exchanged information and social support that helped them maintain better mental wellness.

## 1. Introduction

Rural communities in Australia are heterogenous places; however, a common disadvantage that many rural communities face is limited access to mental health services and support [1,2]. Rural communities face a range of challenges with mental health service provision including service shortages, long waiting lists, and financial and geographic barriers that create difficulties for people seeking to access services [3,4]. Online health resources have been proposed as offering solutions that respond to some of the difficulties with accessing place-based mental health services in rural communities. Innovations such as online peer support forums have become new spaces of mental health support presenting opportunities for the emergence of new (online) communities of interest which are not place-bound [5]. This online mode of delivery of services to rural people has expanded as face-to-face services have become centralised in urban areas in Australia and other countries [6,7].

The use of online peer support forums has become a rapidly evolving space in health care provision that is of particular interest to rural health. While studies have observed how people access emotional support and information through online forums [1,8], the present study examines these processes through a close analysis of social capital, with particular attention to knowledge resources and identity resources built and used on forums [9]. A sense of belonging and connecting with other people is linked with mental wellness, and we are interested in the ways in which interactions on the online forums create impacts for forum users. We sought to understand how social capital might be implicated in these forum interactions and analysed interviews from a study conducted with users of three online peer support forums.

In this paper, we investigate forum rural users’ use of online peer support forums and perceptions of how their interactions impact their mental wellness. Drawing on theories of social capital [9,10], we analyse how forum users build and use social capital in the forum to maintain better mental wellness.

### 1.1. Social Capital

Social capital has been an enduring concept of interest in public health research [11]. Much research about mental health in relation to social capital is based on quantitative studies, in which measures of social capital are generally based on a small number of questions [12]. This study uses a qualitative approach to take a more holistic view of social capital and mental health forums. Drawing on Woolcock’s conceptualisation of social capital as “encompassing the norms and networks facilitating collective action for mutual benefit,” [10] (p. 155), Falk and Kilpatrick highlighted the role of knowledge resources and identity resources embedded in social capital networks which network members can activate or mobilise through interactions [9].

In examining the mechanisms of through which social capital is built and accessed, micro interactions have been theorized over time as producing two broad categories of social capital resources: knowledge resources and identity resources [9], as shown in Figure 1. We were interested in considering this theory proposed by Falk and Kilpatrick in relation to interactions such as those that are central to online peer support forums and how they can work to support mental wellness.

Knowledge resources have been defined as the “knowledge of how to get things done, developed through building capacity and getting to know ‘who knows what’” [13] (p. 743). Values and attitudinal attributes of a community are a knowledge resource [9], and the values and attitudes of a community are a ‘social capital’ resource that helps that community work together.

Identity resources are attributes that facilitate the production and reproduction of “identities of self, others and place as a product of various knowledges, skills, values and collective resources” [9] (p. 100). Knowledge and identity resources together contribute to personal autonomy, which can be considered as having the capacity and the information and knowledge for thinking, making decisions and acting independently [14]. It is this shifting identity-formation that facilitates people’s capacity and willingness to act for the benefit of the community, and to act in roles and ways which their previous perceptions of self did not allow.

### 1.2. Rural Mental Health in Australia

As in other countries around the world, rural mental health is of particular concern in Australia [15]. High suicide rates, particularly for males have been associated with lack of anonymity and barriers to accessing services [16]. Research by the Royal Flying Doctor Service [17] (p. 15) outlines several features of rural Australia that may exacerbate mental health issues and contribute to suicide rates, such as:


*Poor availability of, and access to, primary healthcare and hospital services; limited supply of specialist professionals and mental health services, including fewer psychiatrists, psychologists and mental health nurses per head of population; a reluctance to seek help for mental disorders; concerns about stigma; distance and cost associated with travel to access services.*


Navigating mental health in a rural community is “a unique lived experience because of the many barriers associated with seeking and receiving mental health services” [3], including long waiting lists and high out of pocket expenses. Digital health supports such as online peer support forums have become increasingly available to and used by people in rural communities as access to the internet has expanded in rural areas [18]. Online health resources have been proposed as offering a useful complement that respond to some of the difficulties with accessing place-based mental health services in rural communities. While not a replacement for in-person services, online mental health services have nevertheless been increasingly used by people, particularly through the COVID-19 pandemic [19].

### 1.3. Online Peer Support Forums

The online peer support mental health forums that are the focus of this research operate as facilitated peer support services, with users giving and receiving support through their online interactions on the forum. Most, including the forums in this research, are facilitated by nonprofit mental health organisations. Such models of online peer support have been found to deliver considerable mental health benefits [20] through providing emotional and informational support [21], social connectedness, empowerment and helping people cope [2,22].

Forums operate on the basis of anonymity which is designed to protect forum users’ identities and privacy. This supports users’ personal autonomy [14] by enabling users to disclose information about their mental health challenges with a sense of safety. As part of the project being reported on in this paper, Kang and colleagues examined the role of online peer support forums in the social process of fostering resilience among users of an online forum which employed a model of peer support [23]. Reciprocal interactions are central to the forum’s processes, with users posting on the forum and responding to the posts of other users. Shared norms and values are promoted within forums to build a supportive community, and moderators are employed by nonprofits that run the forums to manage and maintain the community values [24]. Over time, there is the expectation that forum users will internalize these values and promulgate a community-based sense of reciprocity [25]. These norms amongst the forum communities “enable cooperative and trustworthy behaviour in the social environment” [25] (p. 1229).

Online peer support forums have a shared vision for supporting people’s mental health so they can become well and stay well. The forums in the present study have appointed peer support volunteers, which we refer to as ‘peer mentors’ [24] as well as moderation processes to help promote and maintain community values. Moderators on the forum mould and meld the culture and ways of operating in the forum to help to share that vision with the community of forum users, stepping in by removing posts where needed to maintain community standards. At times, the forums’ moderation processes involve moderators excluding some users for ongoing breaches of community standards in relation to messages posted on the forum. The maintenance of community standards enables users to trust the forum, knowing they can feel safe there [24]. Additionally, there can be institutional trust [26] in the nationally or widely recognised brands of the organisations that operate such forums. The forums promote conditions for the construction of social capital for people from diverse backgrounds. As an example, Beyond Blue offers “a tailored approach for specific groups such as rural communities; Aboriginal and Torres Strait Islander people; lesbian, gay, bi, trans and intersex (LGBTI) people; and culturally and linguistically diverse (CALD) communities” (www.beyondblue.org.au (accessed on 26 October 2022). This supports forum users to create their own social fabric, as they build social capital, developing personal autonomy and putting it to work for the benefit of their mental health.

## 2. Materials and Methods

This paper reports on an exploratory qualitative study [27] conducted with a sample of online peer support forum users. The research questions this paper addresses are:Do interactions on online peer support forums provide access to social capital resources that can be used for mental wellness outcomes, and what is the nature of those outcomes?How do interactions on online peer support forums build social capital resources that are available for their users?

We draw on research undertaken in an Australian Research Council project (deidentified project number). We focused on the experience of rural users due to the noted challenges of accessing mental health supports in rural areas. We collected forum post data and recruited forum users for interviews from three well-known Australian online peer support mental health forums hosted by well-regarded nonprofit organisations, namely: SANE Australia, ReachOut and Beyond Blue. These organisations were selected as they are national in scope and are used by rural people. To isolate posts made by people living in rural areas, we defined ‘rural’ as all areas outside of ‘Major cities’ according to the Australian Statistical Geography Standard (ASGS) Remoteness Structure [28]. We then plotted forum posts by the XY coordinates of the postcode that forum users volunteered when they registered to use the forums. By mapping posts by postcode over areas we defined as rural, we were then able to isolate all posts made by people living in rural areas for analysis.

All three forums had large number of users during the data collection period for this study. For example, over the time of the study (August 2018–December 2020), Table 1 shows the quantity of new posts over the 29 months of the project. More than 200,000 posts were made Australia-wide and just under 30,000 posts were made by rural users. The forums held large number of past posts, which were available for users to browse, in addition to being able to respond in real time.

In this paper we draw on the interviews conducted with forum users and present perceptions of the respondents, structured according to themes found in the literature using data analysis methods described below.

### 2.1. Data Collection

Before the recruitment of interviewees began, we needed to ensure that forum users consented to their forum data being reused for research as well as the research aims of this study aligned with the values of each organisation [29]. Having navigated these issues with relevant community managers, ethics approval was granted by Swinburne University Human Research Ethics Committee (R/2019/033). Interviewees who use the forums were recruited via an online ‘expression of interest’ forum posts. When completing this form, participants provided their postcode, which was used to validate whether they lived in a rural location. Rural people who use SANE Australia forums had the largest response rate (*n* = 80). However, recruiting ReachOut and Beyond Blue rural forum users was challenging, despite re-posting the expression of interest form multiple times over the course of 6 months. In the end, the sample was SANE: 20 people, ReachOut four; Beyond Blue six (total rural interviewees *n* = 30). Interviews were conducted between May 2021 and June 2022. Due to rural locations, COVID-19 restrictions and respect for users’ anonymity, twenty-four interviews were conducted by phone and six via Zoom. Interviews were semi-structured [30] and asked members to reflect on their experiences with using the forum and how their forum use influences (or not) their daily life in their rural community. All interviews were audio-recorded with consent, and transcribed verbatim.

### 2.2. Data Analysis

Interview data were analysed inductively following a qualitative thematic analysis method [31]. We looked for themes related to capacity of the forum generally for building strength, coping-building and personal thriving, including social capital-related themes such as social connection, shared values, norms and attitudes and willingness to share information with other users. We were open to new themes that relate to personal thriving in rural communities. For reliability, four researchers initially read the post and interview data, noting themes and other ideas independently. Agreement was then reached on an initial codebook outlining what would be included/excluded for each thematic code. Following this, interview transcripts were coded systematically and independently by three researchers, using NVivo (Release 1.5, QSR International, Burlington, Burlington, MA, USA). Disagreements during the coding process were minor, and each was discussed until a consensus decision was reached. To protect forum users’ anonymity, we use gender-neutral pseudonyms and nongendered pronouns when discussing interview data.

## 3. Results

Rurality was evident in the data as part of the context in which users lived. The rural context had certain qualities to it, particularly a sense of being surveilled in small communities and shortage of mental health services, which often led users to seek out alternative supports such as online peer support forums.


*There may only be a handful of people that may not necessarily want to talk out loud, whereas with the forum, you’ve got that privacy of your personality… Like, because we’re a small town, everybody would know what you were doing if you were going to a group or something like, I guess, drug rehabilitation or Alcoholics Anonymous or something.*
—Kai


*It’s challenging, like, most of last year, there was one permanent GP [general practitioner] in town. And they said, oh, we don’t really do mental health and definitely not as bad as you’ve got or whatever, yeah. We don’t feel comfortable treating you. You need to see someone else. Like there’s no one… The forum’s always been a big support. They’re just always there when the other things aren’t there or not working out.*
—Ray

We found evidence that the forum facilitates the production of social capital that users can access during interactions and use to maintain better mental wellness. Further, people simultaneously used and built social capital through interactions on the online forums as the examples below demonstrate. This section is organised according to three types of outcomes of social capital use we identified in our data: accessing knowledge and information, receiving timely social support, and building confidence and taking steps toward wellness. The Discussion section considers the process through which social capital is built as it is used.

### 3.1. Access to (and Giving) Knowledge and Information

Rural users said they used the forum to find information about mental health conditions, how other people have managed their mental health, or to help them navigate the mental health system. Initially, some users did not anticipate social interaction, rather they expected a one-way transaction, reading forum posts, but not responding:


*I actually registered quite a long time ago, I guess it was when I, myself, you know, at a low time. And to more view rather than write, to see other people’s experiences and to assist my understanding of my own condition. And to see how other people responded and get through their struggles.*
—Alex

Forum users said they trusted the information given by other users who had lived experience of medications, treatments and episodes of mental ill-health. The experience of SANE forum user Drew showed that by interacting and asking questions instead of just reading, they could access information tailored to their own needs:


*People talk about medications. People talk about treatments. People—that’s the thing—it’s really wonderful that you can tap into other people’s experience and knowledge. And that’s the wonderful thing about it. Because you can go, oh, haven’t heard of that. And that may be just the one thing that you need to keep your head above water. It’s a wonderful, wonderful thing. Because it’s not about all these professionals talking to you. You’re talking to likeminded people and people who have tried different things that maybe you haven’t.*
—Drew

Beyond Blue user Taylor spoke of receiving suggestions from peers on the forum that led them to seek help from a GP and obtaining a mental health plan, which opens up access to other mental health services in Australia:


*When I first started [using the Beyond Blue forum], I never left the house. I wouldn’t leave the house at all. Then with help from suggestions people made on the forum, I went to the GP and got a mental health care plan.*
—Taylor

Participants indicated that over time they came to share knowledge and information on the forum for the benefit of others. As a result of that reciprocity, the forum provided access to expertise—in the form of the lived experience of peers on the forum—while developing a sense of the value of their own expertise by offering support to others. That expertise helped people, as Beyond Blue user Jordan explained:

*I think it really makes me feel a lot better having the users who are on there, quite constantly respond to a question that I have. Because I can see, they’ve been on there for 10 years and 4000 posts. They’re quite active, so hopefully, they know what they’re talking about. And I couldn’t say they’ve got anything deeply wrong at all to date*.—Jordan

Jordan explained how this form of expertise helped with managing mental health challenges in times of need:


*When you’re going through anxiety and depression… starting to catastrophize everything and worry about everything, you can’t get out of that mindset… by yourself… so if there’s something that you can do, like going on to a forum and asking a question, say, I’m having this problem, what can I do? The answer that comes back is a way to sort of break out of that cycle. So rather than going to see a psychologist or a psychiatrist or something like that, you might be able to get some quick relief from somebody else’s experience.*
—Jordan

Jordan further indicated that the knowledge gained from others was something that over time they had been able to give back to the online community:


*I’m on this [forum] sort of semi regularly, either to help myself out, or on the off chance that I might be able to help somebody out by giving them things that I’ve learned along the way… I think it sort of sort of validates what I’ve learned can be applied to somebody else going through the same thing. When I started on the forum, I didn’t know a thing. So I had to get that perspective from other people. And now looking at it from a different perspective, I can then pass that on to somebody else.*
—Jordan

Through accessing the wealth of knowledge available on the forum, Jordan gained personal benefits and was also able to share that helpful information with other users.

### 3.2. Receiving (and Giving) Timely Social Support

Rural forum users spoke about the social bonds they had formed with other users of the forums, which over time transformed the forums into spaces they could count on for mental health support. Networks were a key social capital resource for accessing support. Users described how being on the forum and interacting with other people who have experienced similar challenges helped to overcome feelings of loneliness and isolation.


*Depression makes you feel like you’re alone. And it makes you feel like you’re the only one going through what you’re going through… Whereas, you get onto the SANE forum and there’s so many people who are going through the same thing as you. It takes away that feeling of isolation. It takes away that feeling of oh, there’s no one there for me. I just feel so alone.*
—Drew


*There’s a sense of unity there and love, even. And you feel supported and cared about. You feel a bit, yeah, the opposite of not being judged. What would you call that? Accepted.*
—Kerry

In these comments Drew and Kerry indicated that they gained a sense of being there for other people who cared about them. While the forum users are anonymous to each other, the space nevertheless was one of community as Lee explained:


*There’s a shared social space and there’s individual threads where, like, a handful of people might frequent just to check in on each other. But overall, I think yeah, it’s definitely a community kind of environment.*
—Lee

ReachOut user Ali explained how the social networks and bonding social capital of the forums helped support their mental health through interactions that were meaningful because they were personalised:


*When I’m using the forums, I can speak about my particular situation, if it’s something really specific that’s bothering me, or, you know, specific symptoms, I can type that in, I get a response from a real person [which] is very positive. So I have found that really good… I think the feedback’s personal… you kind of get validated. It’s not so much I’ve just kind of sat on the sidelines and watched. I guess also it gets me out of that self feedback loop, kind of gets another perspective. And I get out of my head a little bit.*
—Ali

Trust is an important social capital resource and for users including Riley, it was integral to the confidence they had that the forum would be there for them when they needed support. Riley spoke of trusting the forum community more than the local community. This was particularly key at the time because Riley’s support worker had just been defunded.


*They just said we have a long waiting list and we have to take some of the older clients off it, so it was taken away.*
—Riley

Experiences such as this erode trust in physical community based mental health supports. On the other hand, Riley said they had used the Beyond Blue forum for five years and in that time had built a small community of people who could be counted on to respond during difficult times:


*If I’m having a bad day I can go onto my (BB) thread and post about what’s going on and I know there’s about half a dozen or so people who support me on there.*
—Riley

That support was reciprocated by Riley, who also sent messages to those same people if any of them posted messages signaling distress, or if they had been missing from the forum for a while.


*We look out for each other. If we haven’t heard from someone for a while, you know you just put a little message there ‘hope you’re doing okay’, that sort of thing.*
—Riley

For Riley this small group of people was their community:


*I don’t often go out, my family don’t live near me, so my BB group is my community. Some people think it doesn’t really count because it’s just writing messages to each other, but it does count. You can feel that they care. And they’re there for me.*
—Riley

SANE user Ash spoke of benefitting from receiving and giving support on the forum:


*Once the kids are tucked in bed, and you know, at night-time when it’s the hardest, there’s always someone online that you could just unload to. And then, obviously reciprocate with other people. They unload and… you’ve got the ability to help somebody else and provide some reciprocal support also. It takes the focus off what you’re actually experiencing right now, if that makes sense.*
—Ash

As the comment above demonstrates, the reciprocity of receiving support and supporting others produced multiple benefits for Ash.

Some users indicated that they used the forum to foster friendships. They built bonding social capital with others who they perceived shared similar experiences and who understood them. In contrast, some users indicated that they did not form friendships and social connections on the forum. Jordan commented:


*As far as friendship, I wouldn’t have said that something that’s come out of it [the forum], more so anonymous acquaintance.*
—Jordan

Billy was also reluctant to describe the forum as a place for generating friendships:


*I have a regular number of people with whom I interact. I am not sure that I would use the word friendship on the forum, in the same way as I would use friendship. In everyday life, it’s a different type of relationship.*
—Billy

While participants mostly reported positive experiences of social interactions on the forum, Ash also recounted a negative experience, where a person they had formed a social connection with on the forum, “disappeared” due to the moderation process:


*There was a lady who was frequently on the forum but was clearly unwell and even suicidal. And she just disappeared. And about two years later, there was a brief reappearance where she said that her ban’s over. And then she disappeared again.*
—Ash

The ban mentioned by Ash refers to a process employed by forum moderators whereby users who do not observe the community ‘rules’ can be excluded for periods of time. Ash spoke of concern for the person excluded when no explanation was given about their disappearance:


*I genuinely believed that she’d gone ahead and ended her life.*
—Ash

### 3.3. Building Confidence and Taking Steps toward Wellness

In addition to finding the forum to be a place of social support, as well as information and knowledge about mental wellness, many found the forum offered a means of building self-confidence to improve their situation.


*Once you’re on the forums and you get to find out more about other supports available… it opens your mind up to understanding there’s more knowledge, the more knowledge you have, the more you can understand, and I guess, to help yourself get better.*
—Alex

The bonding social capital that was evident in the friendships and support that Riley gained from the forum built a sense of self-confidence that spilled over into their physical community through volunteering as a step towards helping with depression:


*I read in some of the information that people posted about what can help with depression is that volunteering might help. So I thought, okay. I started to volunteer two days a week at [Charity]. And that’s something I never would have thought of doing, if it wasn’t for someone suggesting it on the forum.*
—Riley

Others described how the forum helped them make steps toward wellness. The forum helped Keelan to feel less alone and be aware that ups and downs are expected:


*It helps to know that there are other people out there who’ve been struggling as long as I have. And it’s okay not to be cured yet. It’s a day-by-day thing, you try to get through each day the best you can. And you’re not failing because you’re having a bad day.*
—Keelan

Drew spoke about building the confidence to focus on mental health in the way that felt personally right:


*[On the forum] you haven’t got the professional showing you their list of things of what they think is best for you. It’s sort of left in your own hands, which gives you a sense of purpose as well. You know, you, okay, well, I’m driving my life, which is so important for people with depression. I’m driving my life and I’m going to give this a go. There’s a proactive nuance to it. And yeah, I think that that’s really important.*
—Drew

The anonymity of the forum enabled ReachOut user Zan to choose how they would like to present themself in forum interactions, which supported their sense of identity:


*I think it’s really helpful because you can pick sort of what you want others to know about you. And it’s a safe environment. Because in real life, if that was to happen, you would choose what you wanted to say. And then social media comes into play and like, friends of friends and family and everything, everything gets very community communicated and you don’t really have a choice in like, who you would like to be anymore.*
—Zan

For some forum users, such as Lee, this confidence facilitated through the forum was a resource that enabled them to take action at high levels to benefit their community. Through being a ‘peer mentor’ on the Beyond Blue forum Lee explained that they had built a rich bank of lived experience knowledge to share and had been invited to provide input into government mental health policy:


*I’m about to pop up to [City] to take part in the lived experience forum which is hopefully going to give the [policy review] some fresh ideas on how the experience can be incorporated, not just suicidality, but all forms of crisis and mental health issues. And if I didn’t have the daily interaction on the forum, and I do, I am on it every day, then I wouldn’t be in a position to say what’s needed.*
—Lee

## 4. Discussion

Addressing the first research question, the findings above show that interactions on online peer support forums provide access to social capital resources that can be used for mental wellness outcomes, including receiving social support when needed; receiving information to navigate the experience of mental ill-health and the mental health system to access services; and building confidence and taking steps toward maintaining better mental wellness.

Findings also showed how social capital was built as people accessed and used it through forum interactions. Riley above is a good example of how interactions on an online peer support forum can assist with forming social connection, suggest how to navigate the mental health system, and increase self-confidence sufficiently for Riley to post on the forum to support others. All of these are outcomes of using social capital resources built though forum interactions. This is evidence of “shifting identity-formation in such a way that facilitates people’s agency, willingness or capacity to act for the benefit of the community” [9] (p. 100).

Members of the forums drew on multiple social capital resources from the forums [9,32]. In return, they were prepared to be there and help other users. This suggests that reciprocity [33] may be an important part of how the forum was able to provide the kinds or nature of social capital resources that rural users were able to use to assist them maintain better mental wellness. The remainder of the Discussion addresses the second research question: how do interactions on online peer support forums build social capital resources that are available for their users?

### 4.1. Simultaneous Building and Using of Social Capital

Here, we unpack the findings to expose how forum characteristics give affordances which allow for and facilitate the processes that build social capital resources and make them available to users. We draw on the model of simultaneous building and using of social capital (Figure 1) which depicts knowledge and identity resources which come together in interactions for the benefit of the community and/or its members. We examine the various elements of, firstly, knowledge resources, and secondly, identity resources, which helped forum users to maintain better mental wellness. We consider how these resources come together as social capital resources in forum interactions. We then consider how the characteristics of online peer support forums afford the process of building and use of forum social capital resources. Elements of knowledge and identity resources listed in Figure 1 are bolded in the following two subsections.

#### 4.1.1. Knowledge Resources

A strength of the forum is the people on it, that is, in terms of the number of users and the breadth of experiential knowledge they have. The substantial **network** of peers from rural and metropolitan locations with varied experiences of mental illness and the health system gave users access to timestamped archives of lived experience [3]. This “knowledge of how to get things done” [13] (p. 743) helped rural users to navigate the mental health system and assess what treatment options they wanted to pursue.

Forum users, for example, in response to urgently needed social support, had the **skills and knowledge** to provide support in a way that helped their peer users to get “another perspective” (Ali), “takes away that feeling of isolation” (Drew) and helps them manage through the inevitable ups and downs of mental ill-health (Keelan). Users such as Jordan developed the skills and knowledge as they saw posts of others on the forum and reactions to these.

The forums all had established **precedents, procedures and rules**, community guidelines set by the nonprofit organisations that established and hosted them. Guidelines were largely helpful, for example, employing moderators and designating ‘peer mentors’ who keep posts within forum scope and structures for storing forum posts that help make forum social capital resources available. There were negative consequences, however, when users were banned and, in turn, deprived of access to the forums and their social capital resources, and for other users who worried about what had happened to people who suddenly stopped posting.

The online forums are **communication sites** that are always available, where the rural users interviewed reported there was always someone ready to reply when they needed support. The forums are designed as supportive and caring spaces by host nonprofits, which encourages interaction [26]. An additional feature of the forums as communication sites was the timestamped archive of past posts that could be accessed whenever users wanted to: a temporal dimension not available in face-to-face communication sites.

A mutual interest in mental health and generating wellness contributes to shared norms and values and to the forum communities’ **value/attitudinal attributes** which reinforce the forums as safe caring and sharing spaces [25]. Value/attitudinal attributes underpin users’ willingness to share when things are not going well and ask for help. The helpful nature of posts in response to needs, or just to share experiences that may be useful to others highlights the importance of the value/attitudinal attributes to the forum’s capacity to build social capital resources.

The forums were a space to go to for social support and connection, as well as knowledge and information. Forum users’ lived experiences of mental health conditions and using services provide a large bank of knowledge and skills about how to engage with the mental health system, as well as access to a wide array of networks. The forum values/attitudinal attributes established and guided by host organisations, and reinforced by how users interact on the forum, combine with the bank of knowledge and skills to form a rich set of knowledge resources.

#### 4.1.2. Identity Resources

Identity resources underpin “willingness and ability to act for the greater good” [13] (p. 743). The findings show user identities, such as for Riley and Lee, are developed and re-shaped over time as users interact, begin to know each other, come to trust each other and develop shared values [9]. Being anonymous on the forum not only helped Zan to feel safe to interact on the forum, but anonymity supported their personal autonomy to choose and shape their own identity [14].

Users perceived that forum interactions built their **self-confidence** and enabled many to take steps towards improving their own mental health. Riley gradually built self-confidence to post which spilled over into everyday life, where they went from never leaving the house to volunteering in the local rural community. Lee built sufficient confidence to act for the greater good by providing input in person into government mental health policy review.

Forum users came to share forum **norms, values and attitudes** as they interacted on the forums over time [25]. These shared norms, values and attitudes supported building self-confidence, and are essential for facilitating the actions users described that result in benefits for forum users [10]. Anonymity and clear community guidelines helped reinforce the vision of the forums as safe spaces for social support and connection, and for finding knowledge and information that is relevant and valuable for individual users.

Over time, users developed **trust** in the forum as a safe space [25] that delivered benefits for them [26]. As discussed in the knowledge resources section above, individuals’ trust in the forums underpinned their willingness to draw on their archives of experience to provide benefits for fellow users through interactions.

While many interactions are motivated by the need of the user who initiates them, users’ **commitment to community** is clearly demonstrated by the stream of interactions described in the findings which are deliberately intended to benefit fellow users.

### 4.2. Forum Characteristics

The knowledge and identity resources built and available to rural users on the forums are facilitated by forum characteristics, which can be viewed as institutional affordances and affordances of technology. These affordances encourage users to interact to jointly build a set of knowledge resources and create conditions that over time, also build identity resources. Figure 2 below depicts how the online mental health peer support forums build and share social capital resources to help rural users maintain better mental wellness. Terms bolded in this section are key elements and attributes of the forum which appear in Figure 2.

#### 4.2.1. Institutional Affordances

The forums were established and run by reputable nonprofit organisations, well-known in the Australian mental health arena. These organisations bring **institutional trust** which signals a safe environment for online interaction [26]. This provides a strong foundation for trust built as users’ interactions on the forum appear to be crucial in reshaping identities and developing self-confidence to share and support other users. The organisations bring established values which are the basis for the values of the forums. The organisations set up a set of rules and procedures, which are reinforced through posts by moderators and designated peer mentors. Moderators also implement rules (that are known to users) through intervening to block posts and in extreme cases, banning users. Moderation keeps the forums on topic and true to forum values [24].

The forums offer peer support, a type of service which has proven effective in supporting mental wellness, both face-to-face and online [2,20,21,22]. Institutional trust and reputation foster a large network of users and a bank of past experiences.

#### 4.2.2. Affordances of Technology

Distinct from physical services, the forums are always open and easily accessible to rural users if they can access affordable, reliable internet. The online space is elastic; differing from, physical rural mental health support services, many people can access it simultaneously [2,17]. The online space of the forum affords the opportunity to be anonymous. Anonymity not only reinforces the safety of the forum, as Zan noted above, but it allows people to reshape their identity and move toward an identity as a person who feels more in control and maintaining better mental wellness.

Reciprocity is at the heart of the forums’ success at helping users move toward being, becoming and staying well. Users simultaneously build and use social capital over time as they seek and share support for each other through interactions [23], reinforcing the mental health benefits of forum use. Forum reciprocity, the social capital resources that reside in the forums and the interactions that take place are especially valuable to rural users who have restricted access to mental health services [17] and many of whom have limited supportive social connections [2].

## 5. Conclusions

The study suggests that interactions on online peer support forums do provide access to social capital resources that rural users can use to maintain better mental wellness, in terms of access to information that they can use, better social support and greater self-confidence that assists them take steps toward better mental wellness. Interactions on forums build social capital resources over time as users access the forum through a process that encourages reciprocity and is facilitated by the affordances of technology and the institutional affordances given by nonprofits.

The power of the combination of affordances of technology and institutional affordances that nonprofits bring is especially significant at a time when rural mental health services are struggling to address needs, and in the context of limited depth and breadth of social capital resources available in physical rural communities. We suggest that mental health service design should consider how physical services can work with nonprofits and technologies to fill gaps and complement services for rural people.

A contribution of this study is that it presents the perceptions of the users of the forums, that is, of the people who have certain mental health conditions and want to overcome them. This type of research gives visibility to people, it makes their experiences material in a digitised way, allows us to understand their difficulties and their efforts, the struggles to overcome their illnesses, anxieties or depressions and the importance of peer forums to develop forms of community, support and mutual care, whether urban or rural.

There are some limitations to the study. We interviewed only small numbers of rural users of three Australian online peer support mental health forums. Interviewees had to have sufficient motivation and self-confidence to respond to a project recruitment forum post and agree to be interviewed. It is possible that other rural people with mental health conditions are not attracted to the forums or have tried them and found they did not meet their needs. Many users are readers only—they do not post on the forums—and we have not been able to capture their experiences. Further research is suggested to look into preferences and outcomes of rural people with mental health issues who do not use online peer support forums, and into how nonprofits and physical rural services can work more closely together to facilitate better mental wellness outcomes for rural people.

While this study focused on rural users of online peer support forums because of the noted challenges of accessing mental health supports in rural areas, we do not suggest that the challenges with accessing mental health services are unique to rural people; nor do we suggest that rural people benefit to different degrees from using online peer support forums than do other users. Metropolitan users of online peer support forums may experience similar benefits. Research comparing rural and metropolitan use of online peer support forums is suggested. Similarly, further research could investigate the usefulness of online peer support forums for various demographic groups, such as rural males.

## Figures and Tables

**Figure 1 ijerph-19-15427-f001:**
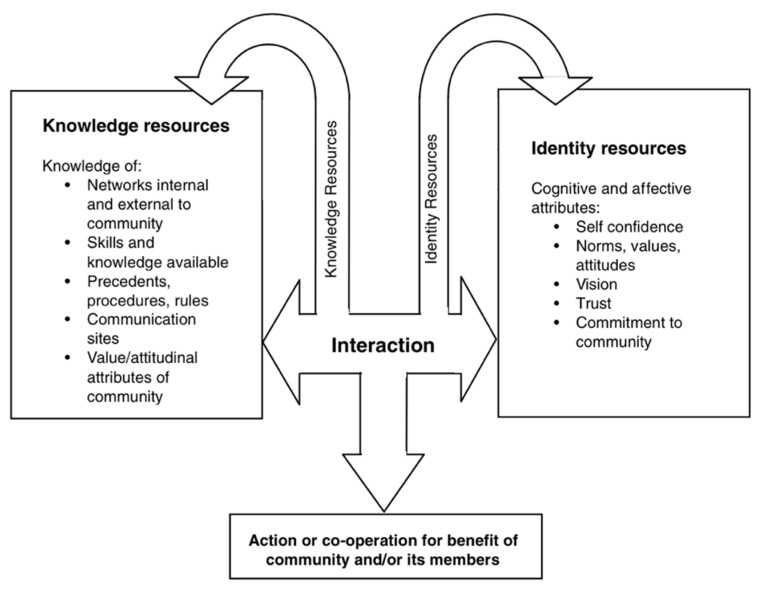
Simultaneous building and using of social capital. Figure from Falk and Kilpatrick, p. 101, reproduced with permission.

**Figure 2 ijerph-19-15427-f002:**
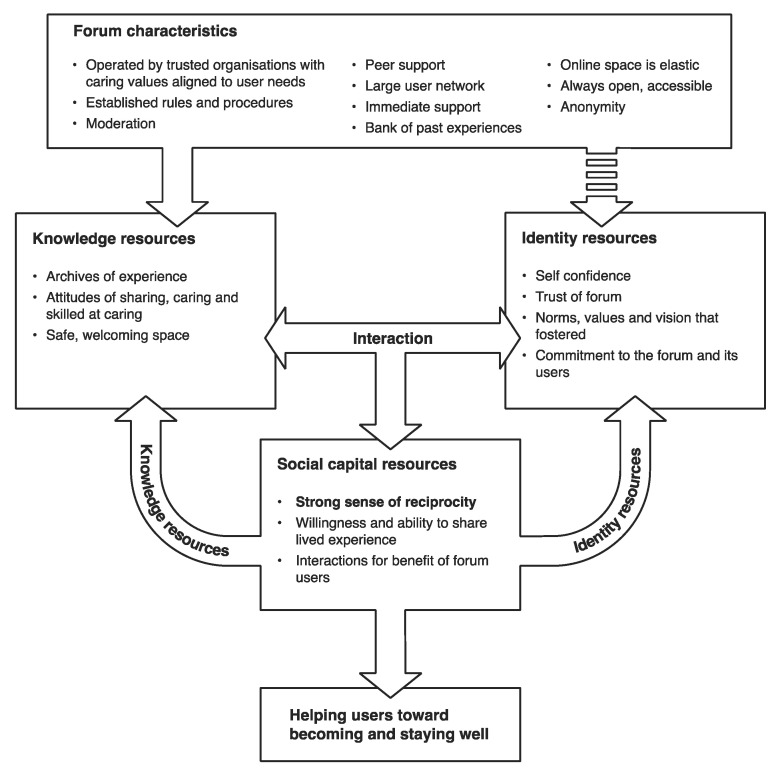
How online mental health peer support forums build and share social capital resources to help rural users maintain better mental wellness.

**Table 1 ijerph-19-15427-t001:** Number of posts and individual authors by nonprofit organisation (August 2018–December 2020).

Forum	Posts	Authors	* Rural Posts	* Rural Authors
SANE	69,950	2322	12,032	251
ReachOut	80,174	1351	11,905	121
Beyond Blue	51,323	7945	5027	684

* Rural includes all areas classified as non-metropolitan [28].

## Data Availability

The datasets used during this study are available from the corresponding author.

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
