# Peer review of "Wellness Impacts of Social Capital Built in Online Peer Support Forums"

_ijerph, 2022, doi:10.3390/ijerph192315427_

Round 1

Reviewer 1 Report

I think that the authors should make some modifications, in my opinion:

1.This article deals with a topic of social and medical interest and importance: the mental health of people living in rural areas. So is the angle from which it works: peer relationships in open discussion forums on the websites of NGOs offering help against depression, anxiety and suicide (I highlight, for review purposes, specifically beyondblue). In my opinion, the article touches on the issues of diversity (functional and territorial) and the personal autonomy in people with mental health difficulties, although it is a concept that is not addressed. It would have been important to discuss the relationship between patient autonomy and social capital.

2. I think it is important to highlight another contribution of the article: it presents the opinion of the users of the forums, that is, of the people who have certain mental health conditions and want to overcome them. This type of research gives visibility to people, allows us to understand their difficulties and their efforts, the struggles to overcome their illnesses, anxieties or depressions and the importance of forums among equals to develop forms of community, support and mutual care, whether urban or rural. However, the specificity of the rural situation is not fully understood because the results may well occur in people from other territorial areas.

3. The patient's autonomy is constructed and therefore its construction should be therapeutically promoted; social capital is also constructed and therefore one of the conclusions could reflect on the promotion of conditions for its construction such as those offered, i.e. beyondblue ("With a tailored approach for specific groups such as rural communities; Aboriginal and Torres Strait Islander people; lesbian, gay, bi, trans and intersex (LGBTI) people; and culturally and linguistically diverse (CALD) communities", www.beyondblue.org.au). This suggestion is made because each individual is creating around their particular needs their own social fabric, their own capital (their own autonomy) and putting it to work for the benefit of their mental health indicates functionality (lines 413-459).

4. Identity is an important result, but it has a double dimension: the rurality of the chosen population and the personal identity associated with the social capital built around their struggle to cope with mental health problems. The first part does not appear and should appear. The second is more developed but the results indicate that it could be rural or urban people, there is no specificity of "rurality" or "isolation" of the people we worked with. It is not enough to say that they have focused on rural participants in the forums, without identifying the motivations that led them to participate as ruralites.

5. Finally, it should be noted that if the empirical results, as presented, can be found in people who attend the forums in similar mental health circumstances, the specificity of rurality is not clear. The lines 73-93 are not sufficient (1.2. Rural mental health in Australia). It seems to me that this is an important point to clarify: Does being "rural" promote or prevent the use of forums? Is the rural condition treated in the forums as a problem of discrimination or exclusion? How is gender coping in rural areas? (i.e. in Beyondblue we read: "depression and anxiety in rural men - Beyond Blue, affecting mental health Men What causes anxiety and depression in men? Men in rural and remote areas, Men in rural and remote areas Isolation and difficulty accessing services are, some of the challenges faced by men living in rural and remote communities. For those making a, die by suicide at rates significantly higher than the general population and non-farming rural males"). It seems to me that there is a bias.

Reviewer 2 Report

Dear Authors

The study concerned the impact of social capital on the wellness of respondents using the internet forum. the research conducted among the community living in rural areas, where access to health  care is difficult.

I read this article with great interest, but I have doubts about the methodology. The respondents (forum users) were contacted and then interviews were conducted directly or online. Who was the interviewer? Was it a psychologist? Did all respondents have the same questions? Have standardized research tools been used to assess social well-being and support, or the sense of coherence, etc., since the authors conclude that these variables improve? (,,The study suggests that interactions on online peer support forums do provide access to social capital resources that rural users can use to maintain better mental wellness, in terms of access to information that they can use, better social support and greater self

confidence that assists them take steps toward better mental wellness”). If so, why were the results not presented?

The results in the form of quotes from individual respondents do not constitute data on the basis of which conclusions can be drawn.

In my opinion, this study presents the subjective, unstructured feelings of the respondents. On this basis, it cannot be concluded that the well-being, social support or the sense of coherence has improved - moreover, in order to assess this, these variables should be compared before the intervention, i.e. the use of support groups.

The conclusions are too "strong" for the results obtained.

Reviewer 3 Report

The paper is a well-designed study carried with the use of a qualitative method.

The Authors verified whether and how social capital resources were generated on online peer support mental health forums, and how they were used by rural users to influence mental health outcomes. The design of the study, structure of the text, conclusions are correct in my opinion.

I would suggest one methodical improvement:

Introduction is rather limited, as well as hypothesis (here rather research questions) development. This is due to short references list - only 24 positions. Please add som latest publications to prove the research gap and then develop hypothesis.
